# Efficacy Analysis of In Situ Synthesis of Nanogold via Copper/Iodonium/Amine/Gold System under a Visible Light

**DOI:** 10.3390/polym13224013

**Published:** 2021-11-20

**Authors:** Jui-Teng Lin, Jacques Lalevee, Hsia-Wei Liu

**Affiliations:** 1New Photon Corp., 10F, No. 55, Sect. 3, Xinbei Blvd, Xinzhuang, New Taipei City 242062, Taiwan; jtlin55@gmail.com; 2CNRS, IS2M UMR 7361, Université de Haute-Alsace, F-68100 Mulhouse, France; jacques.lalevee@uha.fr; 3Department of Life Science, Fu Jen Catholic University, New Taipei City 242062, Taiwan

**Keywords:** polymerization kinetics, photoinitiator, free radical polymerization, copper complex, photoredox catalyst, nanogold particles

## Abstract

This article presents, for the first time, the kinetics and the general features of a photopolymerization system (under visible light), copper-complex/Iodonium/triethylamine/gold-chloride (orA/B/N/G), with initial concentrations of A_0_, B_0_, N_0_ and G_0_, based on the proposed mechanism of Tar et al. Analytic formulas were developed to explore the new features, including: (i) both free radical photopolymerization (FRP) efficacy and the production of nanogold (NG), which are proportional to the relative concentration ratios of (A_0_ + B_0_ + N_0_)/G_0_ and may be optimized for maximum efficacy; (ii) the two competing procedures of NG production and the efficacy of FRP, which can be tailored for an optimal system with nanogold in the polymer matrix; (iii) the FRP efficacy, which is contributed by three components given by the excited state of copper complex (T), and the radicals (R and S) produced by iodonium and amine, respectively; (iv) NG production, which is contributed by the coupling of T and radical (S) with gold ion; and (v) NG production, which has a transient state proportional to the light intensity and the concentration ratio A_0_/G_0_) + (N_0_/(K’M_0_), but also a steady-state independent of the light intensity.

## 1. Introduction

Light sources from UV (365 nm) to near-infrared (980 nm) have been used for photopolymerization reactions in many industrial and medical applications such as dental curing, microlithography, stereolithography, microelectronics, and holography [1,2]. A variety of photoresponsive materials such as conjugated polymers, organic dyes, and metal complexes have been reported for additive manufacturing (AM) and 4D bioprinting [3,4]. Both spatial and temporal controlled 3D processes were reported in systems using single and multiple wavelength lights [5,6]. We have previously reported various strategies for improved photopolymerization conversion, in which the efficacy enhancement is achieved by co-initiators and co-additives in multiple components systems [7,8,9,10,11]. 

Besides UV light photopolymerization, visible lights are also used for organic dyes photosensitizers [12,13], in which high extinction coefficients and their long-living excited states enable the photosensitizers to react efficiently with various additives in the photocurable resins. Cost effective and with long-living excited states, copper complexes have been used as a new polymerization approach to initiate the free radical polymerization (FRP) of acrylates or the free radical promoted cationic polymerization (CP) of epoxides, in which acetylacetonate radicals are produced by redox reaction [14,15,16,17]. Recently, using Coumarin as a dual function, photoinitiators for photo-oxidation and photo-reduction in visible light were reported [18,19,20]. The efficiency of the copper complex (G1) based photoinitiating systems of (G1/iodonium salt (Iod)/*N*-vinylcarbazole (NVK) was investigated by Mokbel et al. [16] for FRP and CP using light sources at 375, 395, and 405 nm. 

More recently, our group reported the novel copper photoredox polymerization for in situ synthesis of metal nanoparticles (Tar et al. [21]) in which nanoparticles were embedded to the polymer matrix to enhance some particular properties of the parent matrix, called polymer nanocomposite. Metal complexes (zinc, ruthenium, iridium, and copper) have been used as photoinitiators for FRP and CP [16,17]. These photoinitiators have excellent photochemistry properties such as strong absorption in visible light, long steady-state excited state, and suitable redox potentials. Furthermore, they also provide dual function of the oxidation or the reduction cycle to produce reactive species—e.g., radicals, anions, or cations. Recently, copper complexes have been gaining more attention due to their comparatively low cost. In our previous work, a new copper complex based on a hydrazone ligand was proposed as a photocatalyst for the polymerization of ethylene glycol diacrylate simultaneously with the production of nanogold in a polymer network under exposure to an LED at 419 nm [21]. 

This article will present, for the first time, the kinetics and the general features of a photopolymerization system (under visible light), copper/iodonium/triethylamine/gold chloride, based on our proposed mechanism [21]. Analytic formulas will be developed to analyze and explore new features that have not been investigated in the measured data of Tar et al. [21].

## 2. Methods and Modeling Systems

The theoretical methods used in the present article include the following steps: (i) setup the photochemical kinetic equations based on the scheme proposed by Tar et al. [21]; (ii) solving the rate equations under the so-called quasi-steady-state conditions, and (iii) finding the efficacy of monomer conversion, which is defined by the time integral of the free radicals. Further details are shown as follows.

### 2.1. Photochemical Kinetics 

A specific measured system was reported by Tar et al. [21] for their proposed kinetic (shown in Figure 1) for a 3-component system of G1/Iod/amine, where G1 is a copper complex HLCuCl, amine is triethylamine (TEA), a reducing agents in the formation of gold nanoparticle inside the polymer matrix; and Iod is iodonium salt (oxidizing agent). The photoinitiating (PI) systems are mainly based on G1/TEA/Iodonium salt (0.05/1%/1% *w*/*w*) to the gold chloride (HAuCl_4_), in which 4 wt% was added in a few drops of DMF, and the PI system was dissolved in ethylene glycol diacrylate (EGDA) at 93.95 wt%. 

The reduction of HAuCl_4_ occurs due to a transfer of electrons from the aminoalkyl radical, but also from the excited state of the copper complex (Cu*). The radical amine^0^ produced from the coupling of Cu* and amine is able to abstract hydrogen to generate radicals, which reduces the tetrachloroaurate Au^+3^ to form Au^+2^, which is then reduced to Au^+1^, and further to Au^0^, leading to the formation of nanogold (NG). The interaction of the G1, TEA, and gold chloride was very fast, (within 60 s) under a visible light at 419 nm. After 5 min under irradiation, the absorption of the complex and Au^+3^ entirely disappeared, having an increase in the absorption at the green (532 nm), which corresponds to the surface plasmon resonance (SPR) absorption [21,22,23]. The peak absorption (at 532 nm) of NG in a spherical shape moves toward near infrared (about 810 nm), if the NG is in a rod shape, with a length and width ratio of about 4.0 [24]. We note that the decomposition of the iodonium salt through an electron transfer and the production of aryl radical could also lead to NG. However, it is a secondary reaction, which is ignored in the present modeling.

We used the following short hand notations: A = copper complex, T = excited state, [HLCuCl]*, B = Iod; N is triethylamine (TEA), R = [HLCuCl]^+*^, G = Au^+3,^, G’ = Au^+2^ or Au^+1^. The monomer (M) is EGDA (for FRP conversion). The associated Chart for Figure 1 is shown in Figure 1 (only the key components are shown). As shown by Figure 1, a 3-initiator system (A/B/N) defined by the ground state of initiator A, which is excited to its first-excited state, and a triplet excited state (T), having a quantum yield (q). The triplet state T interacts with initiator [B], leading to regenerating A and producing a radical R. It can also interact with N to produce another radical S. Both radicals (R and S) can interact with the monomer (M) for FRP. Furthermore, T and R can interact with the gold (G) to produce exited-gold (G’), which further couples with T and S, leading to the formation of NG in the polymer matrix. We note that Figure 1 is more general than that of Figure 1, because it can be used in a general 3-component system, A/B/N, having a various initiator (A), or additives (B and N) and in various metal chlorid (such as gold and silver). The present article focuses on the feature of NG production and limits to FRP. The general case with both FRP and CP has been presented elsewhere [19,20].

### 2.2. The Rate Equations 

The kinetic equations for the concentration of each component of the 3-components (A/B/N) system (shown in Figure 1 and Figure 1) are constructed as follows [19,25,26,27].
(1)$$\frac{d\left[A\right]}{\mathrm{dt}}=-\;\mathrm{bI}\left[A\right]\left(1-R_{E}\right),$$
(2)$$\frac{d\left[B\right]}{\mathrm{dt}}=-k_{1}T\left[B\right],$$
(3)$$\frac{\mathrm{dN}}{\mathrm{dt}}=-k_{2}T\left[N\right],$$
(4)$$\frac{\mathrm{dT}}{\mathrm{dt}}=\mathrm{bI}\left[A\right]-(k_{7}+k_{1}\left[B\right]+k_{2}\left[N\right]+k_{3}G+k_{5}G^{\prime }+\mathrm{kM})T,$$
(5)$$\frac{\mathrm{dR}}{\mathrm{dt}}=k_{1}\left[B\right]T-\left(k^{\prime }R+k^{\prime \prime }S+\mathrm{KM}\right)R,$$
(6)$$\frac{\mathrm{dS}}{\mathrm{dt}}=k_{2}\left[N\right]T-\left({\;k}_{4}G+k_{6}G^{\prime }+k^{\prime \prime }R+K^{\prime }M\right)S,$$
(7)$$\frac{\mathrm{dG}}{\mathrm{dt}}=-(k_{3}T\;+{\;k}_{4}S)G,$$
(8)$$\frac{{\mathrm{dG}}^{\prime }}{\mathrm{dt}}=(k_{3}T\;+{\;k}_{4}S)G-\left(k_{5}T+k_{6}S\right)G^{\prime },$$

In Equation (1) R_E_ is the regeneration (REG) term of of the initiator, [A], given by R_E_ = g(k_7_ + k_1_[B]), with g = k_7_ + k_1_[B] + k_2_[N] + k_3_G. b = 83.6a’wq, where w is the light wavelength (in cm) and q is the triplet state T quantum yield; a’ is the mole absorption coefficient, in (1/mM/%) and I (z, t) is the light intensity, in mW/cm^2^. All the rate constants are defined previously [25,27] and they are related by the coupling terms. For examples, k_j_ (with j = 1,2,3) are for the couplings of T and [B], [N], and G, respectively; K is for the coupling radicals R with monomer M (for FRP). In the above kinetics, we include the bimolecular termination [25] given by k’RR coupling in Equation (5), but for analytic formulas, we will keep only the unimolecular coupling term, KM, for FRP.

The monomer conversions for FRP and generation of NG are given by
(9)$$\frac{\;\mathrm{dM}}{\mathrm{dt}}=-\left(\mathrm{kT}+\mathrm{KR}+K^{\prime }S\right)M,$$
(10)$$\frac{d\left[\mathrm{NG}\right]}{\mathrm{dt}}=\left(k_{5}T+k_{6}S\right)G^{\prime },$$

For comprehensive modeling, we will use the so-called quasi-steady state assumption [25,26,27]. The lifetime of the singlet and triplet states of photosensitizer, the triplet state (T), and the radicals (R, S and G’) is short, since they either decay or react with cellular matrix immediately after they are created. Thus, one may set dT/dt = dR/dt = dS/dt = dG’/dt = 0, which gives the quasi-steady-state solutions: T = bIg[A], R = k_1_bIgg’[A] [B]), S = k_2_bIgg”[A] [N]), G’ = (k_3_T + k_4_S)/(k_5_T+k_6_S); with g = 1/(k_7_ + k_1_[B] + k_2_[N] + k_3_G + k_5_G’ + kM), g’ = 1/(k”S + KM), g” = 1/(k_4_G + k_6_G’ + k”R + K’M). 

Under the above quasi-steady-state solutions, we obtain the simplified equations as follows.
(11)$$\frac{d\left[A\right]}{\mathrm{dt}}=-\;\mathrm{bI}\left[A\right]\left(1-R_{E}\right),$$
(12)$$\frac{d\left[B\right]}{\mathrm{dt}}=-k_{1}\mathrm{bIg}\left[A\right]\left[B\right]\;,$$
(13)$$\frac{\mathrm{dN}}{\mathrm{dt}}=-k_{2}\mathrm{bIg}\left[A\right]\left[N\right],\;$$
(14)$$\frac{\mathrm{dG}}{\mathrm{dt}}=-\mathrm{bIg}\left[A\right]\left(k_{3}+k_{2}g^{\prime \prime }\left[N\right]\right)G.$$

The monomer conversions for FRP and generation of NG are given by [27]
(15)$$\frac{\;\mathrm{dM}}{\mathrm{dt}}=-\mathrm{bIg}\left[A\right]\;\left(k+k_{1}{\mathrm{Kg}}^{\prime }\left[B\right]+k_{2}K^{\prime }g^{\prime \prime }\left[N\right]\right)\;M,$$
(16)$$\frac{d\left[\mathrm{NG}\right]}{\mathrm{dt}}=\mathrm{bIg}\left[A\right]\left(k_{5}+k_{6}k_{2}g^{\prime \prime }\left[N\right]\right)$$

## 3. Results and Discussion

A full numerical simulation is required for the solutions of Equations (11)–(16), which will be presented elsewhere. We will focus on comprehensive analysis for special features and the key factors for efficient producing of NG related to the measured data of Tar et al. [22], based on the analytic formulas. 

### 3.1. Analytic Results

Analytic formulas need assumption of strong coupling of T and [B], and KM >> k”S, K’M >> k_4_G such that g = 1/(k_3_G), g’ = 1/(KM), and g” = 1/(K’M). In addition, R_E_ is taken as a mean reduction factor (f’), such that (1–R_E_) = f’ = [1 − k_2_N_0_/(k_3_G_0_)] is time independent, having a value of f’ = 0.5 to 1.0. The first-order solutions of Equation (11) to (15) are found: [A] = A_0_ exp(–dt); with d = f’bI; [B] = B_0_ exp(-H), with H(t) = DE(t), with E(t) = [1 – exp(–dt)]/d]; [N] = N_0_ exp(–H’), H’ = D’(k_2_/k_1_)D, with D = (k_1_/k_3_)d(A_0_/G_0_). Also G(t) = G_0_ – (k_1_/k_3_)dt; G’ = (k_3_/k_5_)[1 + k_4_S/T] = (k_3_/k_5_)[1 + k_4_[N]/(KM), for k_5_T >> k_6_S. Using these approximated solutions, Equations (15) and (16) become
(17)$$\frac{\;\mathrm{dM}}{\mathrm{dt}}=-\mathrm{kF}\left(t\right)M-F\left(k_{1}\left[B\right\}+k_{2}\left[N\right]\right),$$
(18)$$\frac{d\left[\mathrm{NG}\right]}{\mathrm{dt}}=F\left(t\right)\left(k_{5}+\frac{k_{6}k_{2}\left[N\right]}{K^{\prime }M}\right)G^{\prime },$$
where F’(t) = bI[A](k_3_G). 

Solving Equation (17), we obtain the first-order solution, with G(t) = G_0_, and G’ = (k_3_/k_5_).
M = M_0_ exp[−P(t)] + Ho(t),(19)
where P(t) = Q [1− exp(−dt)]/d]; with Q = (k/k_3_)bI(A_0_/G_0_); and Ho(t) is a complex second order term proportional to bIA_0_(k_1_B_0_ + k_2_N_0_/B_0_)/G_0._ We note that P(t) has a transient state value P = Qt, and steady-state value, P = Q/d, which is independent of the light intensity. 

The production of nanogold, [NG], is given by the solution of Equation (18),
[NG](t) = D’[E(t) + Q’E’(t)],(20)
where Q’ = (k_6_k_2_/k_3_)(N_0_/(K’M_0_), E(t) = [1– exp(–dt)]/d]; E’(t) = [1– exp(–d’t)]/d’]; D’ = bI(A_0_/G_0_); d’ = d + (k_1_/k_3_)D, D = (k_1_/k_3_)d(A_0_/G_0_). We note that [NG](t) has a transient state value [NG] = D’(1 + Q’t); and steady-state value, [NG] = D’(1/d + Q’/d’), which is independent to the light intensity. 

### 3.2. General Features and New Findings

As shown by Equations (17)–(20), the following significant features of the A/B/N/Gold, or Cu/Iod/Amine/Gold, system are summarized as follows (referred also to Figure 1 and Figure 1).

(i) The additive [B], or Iod (an oxidizing agent) interacts with copper excited state (T, or Cu*) to produce radical (R) and also to regenerate the initiator, [A] (or Cu), shown by R_E_ = g(k_7_ + k_1_[B]) term in Equation (11). 

(ii) The additive, [N] (or amine), has dual functions: interacting with T to produce additional radical (S, or amine^0^) which leads to FRP; and coupled with G, or Au(3+), to form Au(2+), then G’, or Au(1+), leading to Au(0) and the nanogold (NG). 

(iii) Equations (9) and (17) show that the FRP is contributed by three components: from the coupling of monomer (M) with T, R, and S. The conversion efficacy is proportional to bI(A_0_+ *k_1_* B_0_
*+ k_2_*N_0_)/(k_3_G_0_). which is an increasing function of the absorption coefficient (b) and light intensity (I), and the concentration ratios of (A_0_ + B_0_
*+* N_0_)G_0_. 

(iv) Equations (10) and (18) show that the production of NG is contributed by two components given by the coupling term of (k_5_T + k_6_S)G’ shown by Equation (10). In addition, as shown by Equation (20), [NG](t) has a transient state value [NG] = D’(1 + Q’)t, which is an increasing function of bI[(A_0_/G_0_) + (N_0_/(K’M_0_); and the steady-state value, [NG] = D’(1/d + Q’/d’), which is independent to the light intensity. 

Besides the above-described features, our modeling has explored the follow new findings, which are not observed in the experiment of Tar et al. [21]. 

(a) Both FRP efficacy and the production of NG are proportional to the relative concentration ratios of (A_0_ + B_0_ + N_0_)/G_0_, rather than the individual concentrations. Therefore, the photoinitiating system reported by Tar et al. [21] based on G1/TEA/Iodonium, having (0.05/1%/1% wt) and 4 wt% of gold chlorid (G_0_, or HAuCl_4_), is not optimized. Our modeling predicts that lower initial gold chlorid (G_0_) and/or larger (A_0_ + B_0_ + N_0_), leads to a higher ratio of (A_0_ + B_0_ + N_0_)/G_0_, and therefore higher FRP and larger production of NG. It seems that Tar et al. [21] have used a too high concentration of gold, but too low G1 concentration. In addition, our formula of Equation (19) predicts an optimal value of the concentration ratio (A_0_ + B_0_ + N_0_)/G_0,_ as also predicted by our previous modeling in other systems by Lin et al. [19,28].

(b) We note that [B] plays no role in the production of NG under the first-order solution, but plays a minor role reducing the NG production, as shown by our second-order factor g = (1 − k_1_[B]/G − k_2_[N]/G)/(k_3_G). Furthermore, there is a reduction effect in the FRP efficacy due to the production of NG caused by the reduction of excited state (T) when it couples with gold, Au(+3). On the other hand, higher FRP (or larger K’M term) also reduces the efficacy of NG production, as shown by the factor Q’ = (k_6_k_2_/k_3_)(N_0_/(K’M_0_) in Equation (20). Therefore, one may tailor the ratio of (A_0_, B_0_, N_0_)/G_0_ to achieve maximum NG production, but also the strength of polymer matrix (or higher FRP), which is a competing procedure of NG production.t

## 4. Conclusions

Analytic formulas are developed to explore the kinetics of a G1/ Iodonium/ TEA/gold chlorid (or A/B/N/G) system. We found that the FRP efficacy is governed by the coupling of the excited state of copper complex (T) with the radicals (R and S) produced by iodomium and amine, respectively. NG production has a transient state proportional to the light intensity, whereas it is independent to the light intensity at steady-state. The competing between NG production and efficacy of FRP can be tailored by the relative concentration ratios of (A_0_ + B_0_ + N_0_)/G_0_, which has an optimal value with nanogold in the polymer matrix.

## Data Availability

The data presented in this study are available on request from the corresponding author.

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
