# Peer review of "Efficacy Analysis of In Situ Synthesis of Nanogold via Copper/Iodonium/Amine/Gold System under a Visible Light"

_polymers, 2021, doi:10.3390/polym13224013_

Round 1

Reviewer 1 Report

See the attachement.

Author Response

see file

Reviewer 2 Report

The article: “Efficacy analysis of in situ synthesis of nanogold via copper/iodonium/amine/gold system under a visible light” by Jui-Teng Lin et al. describes the kinetics and the general features of a photopolymerization system (named as G1/Iodonium/TEA/Gold chloride) under visible light. This study is based on a previous article by Tara et al. (Ref. 21). Although the subject of the article is interesting and valuable, the presentation of this study is not at a good level. The reader is confused with the structure of the manuscript and cannot easily follow the authors’ statements. Unfortunately, in this form is not suitable for publication in Polymers. A serious revision is needed before re-submitting it. Some comments are addressed below.

  1. Line 31-34: The authors repeat the same sentence as above. In general, the introduction part needs serious revision and editing.
  2. In Scheme 1, what are r1, r2, etc.. refer to?
  3. Line 87-90: The sentence has to be revised, the reader cannot follow the authors.
  4. Line 108-111: Where is Scheme 6 refer to?
  5. The Methods and modeling systems seems more like results and discussion section. The authors do not describe the methods they used clearly.
  6. Conclusions and Abstract are almost identical. The authors should revise them.

Author Response

see file

Round 2

Reviewer 2 Report

The authors have answered to the comments provided by the reviewer and revised their manuscript.

Now it can be accepted for publication to Polymers.